# Associative Learning of Quantitative Mechanosensory Stimuli in Honeybees

**DOI:** 10.3390/insects15020094

**Published:** 2024-02-01

**Authors:** Heather Strelevitz, Ettore Tiraboschi, Albrecht Haase

**Affiliations:** 1Center for Mind/Brain Sciences (CIMeC), University of Trento, Piazza Manifattura 1, 38068 Rovereto, Italy; heather.strelevitz@unitn.it (H.S.); trbttr@gmail.com (E.T.); 2Department of Physics, University of Trento, 38123 Povo, Italy

**Keywords:** proboscis extension, honeybee, mechanosensation, learning, decision making, automation

## Abstract

**Simple Summary:**

One major challenge when using animal models to study cognition is that we cannot ask them what they are thinking or how they are feeling; instead, we measure the animal’s behavior. For honeybees, the extension of the proboscis (their tongue-like structure) occurs when they are presented with a sucrose solution, and they can be trained to associate a neutral cue—for example, an odor—with the occurrence of this food reward, eventually extending the proboscis when presented with the neutral cue rather than the food cue. Thus, the proboscis extension response (PER) is useful for exploring honeybees’ sensory perception, learning, and memory. In this study, we tested a new stimulus, namely different speeds of air flow, to investigate whether honeybees were able to associate this cue with the reward. Additionally, we designed a new system for performing PER experiments wherein the stimulus delivery and analyses are entirely automated, rather than performed manually. Using this enhanced method, we found that honeybees succeeded when a lower air flux was rewarded, but not when a higher air flux was rewarded. These results add to our knowledge of stimulus intensity encoding, while our improved PER system will offer technical advantages for such experiments in the future.

**Abstract:**

The proboscis extension response (PER) has been widely used to evaluate honeybees’ (*Apis mellifera*) learning and memory abilities, typically by using odors and visual cues for the conditioned stimuli. Here we asked whether honeybees could learn to distinguish between different magnitudes of the same type of stimulus, given as two speeds of air flux. By taking advantage of a novel automated system for administering PER experiments, we determined that the bees were highly successful when the lower air flux was rewarded and less successful when the higher flux was rewarded. Importantly, since our method includes AI-assisted analysis, we were able to consider subthreshold responses at a high temporal resolution; this analysis revealed patterns of rapid generalization and slowly acquired discrimination between the rewarded and unrewarded stimuli, as well as indications that the high air flux may have been mildly aversive. The learning curve for these mechanosensory stimuli, at least when the lower flux is rewarded, more closely mimics prior data from olfactory PER studies rather than visual ones, possibly in agreement with recent findings that the insect olfactory system is also sensitive to mechanosensory information. This work demonstrates a new modality to be used in PER experiments and lays the foundation for deeper exploration of honeybee cognitive processes when posed with complex learning challenges.

## 1. Introduction

Learning and decision making are core features which allow an animal to survive in a stimulus-rich and dynamic environment. The cognitive capacities necessary to support these abilities are present not only in what we consider to be higher-order mammals, but also in organisms as far down the phylogenetic tree as the tiny invertebrate *Caenorhabditis elegans* [1]. The honeybee (*Apis mellifera*) is a particularly suitable model for the study of learning and decision making considering their skill in foraging: a high-stakes, fast-paced challenge which the average adult bee performs with remarkable success. In a field of various floral signals, a honeybee must choose the targets with the highest chance of reward and must accomplish this as quickly as possible. Underlying the behavior are complex interactions between sensory perception, sensory processing, memory, and motor systems, not to mention a continuous learning process. The mechanisms by which the honeybee efficiently optimizes this task has intrigued researchers for decades.

Here, we aimed to investigate these topics with new improvements to an old method. Since its introduction in 1957 [2], the proboscis extension response (PER) has been thoroughly characterized [3,4] and used with great success to probe the bases of insect learning through a classical conditioning paradigm which is both simple and versatile [5]. Within just a few trials, a harnessed honeybee will learn the association between a conditioned stimulus (CS) and an appetitive unconditioned stimulus (US), usually sucrose. Experimenting with PER takes advantage of the bees’ innate reflex to extend the proboscis when the antennae, tarsi, or mandibles come into contact with sucrose [5], which in healthy bees is a robust and reliable phenomenon. Since associative learning is conserved across modalities, PER experiments have been performed with various conditioned stimuli including olfactory, visual, and tactile stimuli [6]. Some of these are more effective than others: for example, experiments with visual stimuli [7,8,9] never reach as high of a success rate as olfactory experiments [5,6]. Consequently, olfaction has been the most popular CS for PER conditioning.

Typically, this experiment requires the researcher to manually provide some or all stimuli: the sucrose solution would be supplied to the bee from a syringe, and the odor would be delivered from syringes containing filter paper imbued with the odorant. In recent years, odor delivery has been more and more automated and is controlled with custom-built olfactometers which can provide the stimulus with precise timing and duration [10,11,12]. This would be followed by recording the observed dichotomous behavioral response, which quantifies whether or not the bee extends the proboscis to the CS before the presentation of the US [13]. This protocol allows for the investigation of both sensory perception and the abilities of learning and memory and is widely used due to its ease of implementation and low-cost equipment [5]. However, there are a variety of protocol details which are known to cause variation in the PER acquisition, and are difficult to control manually: stimulus intensity, air flow, and timing of stimulus presentation, to name a few [14]. Variations among these factors may reduce the validity of comparing data between different labs or even within the same lab between experiments run by different researchers. Further, the binary classification of behavioral output—often necessary for a manual experiment—is only one piece of the information available and thus limits the scope of our biological comprehension.

In recent years, several efforts have been made to implement automated tracking of honeybee head parts to facilitate experiments and to obtain quantitative information on the bees’ responses to stimuli. Many of the efforts concentrated on tracking antennal motion. An initial attempt used paint marking of antenna tips, which facilitated their identification on camera images [15]. This showed how odor conditioning influenced the antennal motion pattern. Another study with markerless tracking of the antennae and proboscis tip used the large contrast between the tip and the background, which was detected via an algorithm that followed pixel-by-pixel average color changes [16]. Currently, the gold standard in animal tracking is the deep learning python toolbox DeepLabCut [17]. The strategy here is transfer learning from a very deep neural network pretrained on huge object recognition datasets. Antennal tracking experiments have begun to take advantage of this tool [18], again analyzing antennal motion in response to odor stimuli.

Efforts towards fully automatized conditioning have also been made before. One approach performed automatic analysis paired with manual conditioning using Bayesian-based algorithms to identify the motion of antennae, mandibles, and proboscis [19]. Another study implemented a fully automated approach for aversive conditioning within a walking arena, where the walking motion of the bees could be continuously monitored, and electric shocks could be provided via the walking surface [20]. In the presented work, we addressed both problems: the complete automation of the administration of both stimuli (CS and US) in a fully customizable sequence to multiple subjects and real-time response detection supported with machine learning classification of the proboscis extension. The device was tested in a series of experiments to demonstrate its validity for PER methodology, as well as to assess a novel type of conditioned stimulus, mechanosensory information through air flow.

Like many insects, honeybees are exquisitely tuned to perceiving changes in air flow to aid with flight navigation, which is integrated with visual information for optimal performance [21]. As for most flying insects, they detect air flow via their antennae, and honeybees have an additional mechanosensory advantage with their notoriously thick body hairs; in fact, honeybees have a unique use for air flow information during their waggle dance [22]. Interestingly, recent studies suggest that the insect olfactory system might not be limited to the processing of odor information but is also sensitive to certain mechanical stimuli. Structural as well as functional evidence has been found at the level of the antennal lobe [23,24,25,26] as well as in higher-order brain centers [27,28]. Thus, mechanosensation was an interesting candidate as a CS for PER experiments; would it be more effective than some other modalities, such as vision, since it seems to be incorporated into what we call the olfactory perception system? We would postulate that the answer is yes. Since mechanosensory air flow is coded more similarly to olfaction than visual information, its potential for use as a CS during PER conditioning will more closely mimic that of olfaction PER experiments in terms of acquisition pattern and overall success rate.

Mechanosensory stimuli have been previously used for conditioning protocols, elucidating honeybees’ high capacity for tactile learning in both operant [29] and PER frameworks [30,31] wherein stimuli were in the form of objects held near to or touching the antennae. Our air flux stimulus is not only an interesting alternative conceptually but also offers several advantages. These prior studies needed to paint the bees’ eyes black to avoid the confound of visual stimuli, which is not only labor intensive but also interferes with the welfare of the bee. Further, the application of the stimulus is limited in its versatility and precision, since the human hand will inevitably fail to provide identical directions and quantities of forces over trials. By customizing our setup to deliver varying speeds of odorless air flux, we eliminate visual confounds and increase our control over the timing and amount of mechanical stimulus.

The objective of this study was twofold: the first was from a technical perspective, which assesses the validity of an automated PER conditioning system, and the second was a more scientific one, which tests a novel mechanosensory CS during PER conditioning using these improved experimental and analytic methods. For the first aim, we tested the system using olfactory PER experiments (which are well-characterized regarding what a typical result should be) to ensure it can reproduce previous results, thus demonstrating that the automated protocol itself contains no confounds. Specifically, we used forward pairing of the stimuli with differential conditioning between two odors, a common protocol. For the second aim, we performed PER experiments using the same paradigm but with a novel CS: speeds of odorless air flux.

## 2. Materials and Methods

### 2.1. Honeybee Maintenance and Handling

No legal regulations apply to experimental research with honeybees. Subjects were maintained in the outdoor colonies of the University of Trento at their site in Rovereto, TN, Italy. Experiments were performed from July to October 2022. Forager bees, identified by the presence of pollen on the hind legs of a bee returning to the hive, were individually collected from two colonies. With this approach we caught only pollen foragers, likely reducing the variation in individual learning capabilities that would be observed for a more random sampling of all bees leaving the hive [32]. Bees were immobilized on ice to allow for the mounting procedure described below. The duration of the contact with ice was minimized, using only the time required for sufficient handling.

### 2.2. Hardware

The device comprises several elements (Figure 1): the revolver, a wheel where the bees are mounted; the feeder, which provides the consumable stimulus as well as a tactile stimulus to the antennae; the camera, which records the behavioral response of the bee; the stimulus generator, a 2-channel olfactometer (this independent module can be replaced by arbitrary stimulus generators such as visual, olfactory, auditory, mechanical, etc.); the controller, an Arduino-based controller (Arduino, Monza, Italy), that drives all the motorized components and if necessary also the module for the stimuli (Appendix A); and the USB-6009 NI DAQ I/O device (National Instruments, Austin, TX, USA), which is the port of communication between the PC running the main program, the Arduino controller, and the actuators (valves, AP-621L-LR3-GPH, Camozzi, Italy).

All the structural components for the revolver were 3D printed and/or machined based on the technical drawings provided (Appendix A). We used PETG (glycolized polyester), and TPU (Thermoplastic Polyurethane) for 3D printing, and plexiglass for machining. A scheme of the full assembly is provided (Appendix A). All electronic components are listed in Appendix A, and their wiring is shown in Appendix A. The wired Arduino and USB NI DAQ ports shown in this figure correspond to those addressed in the provided software V1.0 (https://github.com/NeurophysicsTrento/Automatized-PER (accessed on 22 December 2023)).

### 2.3. Rotor and Feeder

The device is based on a rotating wheel, the revolver, where 12 bees are loaded along the circumference facing outward (Figure 1B). The revolver positions a single bee in front of the feeder and the stimulus generator. After each trial, the motor spins the revolver to place the next bee into the experimental position. The correct positioning is achieved with a rotary encoder that determines when to interrupt the rotation. A servo motor rotates the feeder (Figure 1C) to apply the US and reward as seen in Appendix A. Further conditioning information is provided in Specific Protocols. The covered revolver forms isolated compartments for each bee to reduce the possibility of experimental stimuli reaching bees which are not in the test position. In the case of olfactory conditioning, the CS odors are removed by means of a suction tube attached below the revolver (Figure 1A). During the trial, a camera records a video of the bee from above, which is later used for behavior classification and data extraction (Appendix A).

### 2.4. Odor Stimulation

Odor stimuli are delivered using a custom-made 2-channel olfactometer. Briefly, a fluxometer feeds air through glass vials containing either an odor diluted in mineral oil or only the solvent. A fast 3-way solenoid valve selects which channel is opened. By default, the solvent channel is continuously streamed through a nozzle aimed at the head of the bee in the experimental arena. The CS odor stimulus is then delivered by activating a solenoid valve. The trigger is sent from the main program through the USB NI DAQ board to the circuit that powers the valve (in this case, a Darlington array chip (ULN2003)) (Appendix A). The appropriate voltage depends on the valve; in our case, it was 12 V for activation of the solenoid and 5 V for holding. Due to the short period of the stimulus, a VCC of 12 V can be used throughout the whole stimulus.

### 2.5. Mechanosensory Stimulation

In order to control the air stream intensity, we used a voltage-regulated valve, such that the air flux is controlled by varying the voltage. The necessary current for operating the valve is provided again through a channel of the ULN2003 chip. The gate of a channel on the ULN2003 must be connected to a Pulse-width modulation (PWM) pin on the Arduino and the voltage-regulated valve must be connected to the ULN2003 as the odor valve (Appendix A). For the experiments presented here, a separate Arduino was used for controlling this valve simply to obtain higher modularity of the system. A calibration of the valve is necessary in order to define its working range, which is based on the input air pressure. In this case, the Arduino is driven with digital inputs received by the main program through the USB NI DAQ board. The flux is measured with a thermo-anemometer Testo 405i, (Testo, Settimo Milanese, Italy).

### 2.6. Software

In order to execute an experiment, the device requires three layers of software. The main program is written in Matlab (R2019b, Mathworks, Natick, MA, USA) and it oversees the entire device’s functions according to the experimental protocols, as well as analyzing the resulting data. The CNN model is a convolutional neural network that identifies the behavioral response of the recorded bee. It has been generated and trained using TensorFlow and the Keras framework. The network is then imported into the Matlab environment for live classification of bee responses. The hardware controller code is written using the Arduino IDE and is meant to control all the motors and actuators for stimulus controls (valves, solenoids, speakers, LEDs, etc.). It can be adapted for different experimental protocols since it controls the synchronization, timing, and type of stimuli.

All the codes are commented with user instructions and can be found at https://github.com/NeurophysicsTrento/Automatized-PER (accessed on 12 January 2024). In the case of the CNN model, the datasets used for the training, and the trained model itself, are provided along with the Matlab code used to manually create labeled datasets from recordings of PER experiments. The software functions are summarized as follows.

### 2.7. Main Code

The main code, written in Matlab, oversees the entire PER experiment. It initializes the USB NI DAQ port for communicating with the hardware; it initializes the camera and controls the video recording and analysis; it defines the exact protocol for the PER conditioning, such as the sequence and timing of all stimuli (PER_Protocol.m); and it saves the final dataset as a .mat file which contains all the parameters of the experiments, the recordings, and the classifications from the AI model. The main analysis script (analyse_PER.m) then uses this file to perform analyses and visualizations of the experimental data. Also, it includes functions to extract important parameters, like the classical learning rate, and contains visualization tools to help summarize the outcome of the experiment and to highlight behavioral features. It then allows a revision of the experimental data before finalizing the analyses.

### 2.8. Convolutional Neural Network

The CNN performs an automated classification of the behavioral responses of the bees. During the trial period, a video of the subject is recorded and automatically fed to an AI model for classification. The classification is based on the extension of the proboscis: the response is defined as “licking” if the end of the proboscis is extended beyond the mandibles of the bee or “rest” otherwise (Appendix A, insets). When the proboscis tip is in close proximity to the mandibles’ edge the classifier will be less accurate, since the manual classification of those frames will not always be scored in exactly the same way between different videos and experimenters. The output of the model is the probability that the proboscis is extended, a value between 0 and 1 that is assigned to each frame. Appendix A shows an example of an experimental trial labeled with the classifier output. The model is a convolutional neural network, generated using Tensorflow and the Keras framework, which takes inputs as frames of size 100 × 100 pixels. The provided Jupyter notebook script (Jupyter project, USA) (PER_CNN.ipynb) can be used to train and retrain a model. In general, the model requires a training session whenever new imaging conditions are met. A good approach is to train the model on videos that are heavily misclassified; for example, if the ambient lighting of the recording has changed, the classifier may perform less accurately, and those recordings should be used for retraining. For the experiments reported here, the model was retrained four times until it was robust enough that retraining was no longer required. The user can create training datasets by means of a Matlab script (movie_Labeler.m) which serves as an interface to manually classify each frame from a video of a bee trial. Usually, for a video of 300 frames, this process requires no more than 30 s (see instructions included in the script). Information about the architecture of the model can be retrieved from the PER_CNN.ipynb file or after loading the model into Matlab.

### 2.9. Hardware Controller Code

This code is written in C using the Arduino IDE (Arduino, Monza, Italy). The provided Arduino IDE code (PER_device.ino) controls the movement of the revolver by activating the stepper motor, and it controls the feeder by activating the servo motor. All the pin assignments must match the wiring of the Arduino (Appendix A). The code defines how long the feeder will deliver the consumable stimuli to the bee or how to move the feeder if the tactile stimulus of the antennae should be excluded. The code for controlling the variable flux valve for the mechanical stimulus is in the Arduino IDE file PER_AIR_Flux.ino. The valve was connected to a PWM pin on a separate Arduino boardwhich receives the input signal from the USB NI DAQ port.

### 2.10. Specific Protocols

Each of the below experiments followed the steps listed in the generic experimental protocol, along with the sequence and timing of stimuli as depicted in Figure 2A, the details of which were optimized throughout preliminary experiments by referring to well-established literature [5,13] and careful observation of behavior (for example, a high rate of proboscis extension prior to CS onset meant that the familiarization time needed to be increased). All subjects received the same stimulus on each trial, and the first trial was always with the unpaired stimulus to ensure there were no confounding variables present since the bees’ behavior should not change (for example, if the bees extended the proboscis during this trial, there was likely some sucrose contamination on the setup). The remaining trials were randomized between paired and unpaired. All bees experienced all trials, paired and unpaired, to control for odor-specific biases. Thus, the provided *n* for each experiment is the total number of subjects.

The sucrose solution was 25% (*w*/*w*) (0.9 mol/L) of granulated white sugar in distilled water. This was changed often—every few days—and stored in the 4 degrees lab refrigerator to reduce bacterial growth. This concentration was determined to be an efficient balance between too high (which causes the bees to more quickly become satiated and lose their motivation to participate in the task) or too low (which may not be motivating enough in the first place). It is a lower concentration than is normally used for PER protocols [13], to avoid satiation due to our lengthier mechanosensory experiments, and did not result in any diminished performance for the shorter olfactory experiments.

To condition subjects on paired trials, the feeder rotates (from the right to the center, from the point of view of the bee) to touch the antennae with a sucrose-soaked toothpick and present the sucrose solution for feeding. Both are required to trigger PER association, since without the initial stimulus the bee might not perceive the sucrose in the feeding vessel. The left feeding vessel is kept empty to serve as a visual control such that the two sides of the feeder look identical. Thus, on the unpaired trials, the feeder rotates from the left so that the empty feeding vessel was presented to the subject.

### 2.11. Olfactory Conditioning

The two odors used for the CS+ and CS− were 1-hexanol and 1-nonanol, respectively. These have been shown to be effective for PER experiments and are well-distinguished from each other [33]. After initial experiments confirmed that the results were invariant to the order of the specific odors—in other words, the honeybees’ performance did not change when 1-nonanol was the CS+ as opposed to 1-hexanol—this detail was kept constant for simplicity. Previous works [33] have also demonstrated the similar efficacy of these two odors. Odors were diluted 1:100 in mineral oil and changed regularly (approximately every two weeks). There were 10 trials in total, with 5 of each CS. There were *n* = 24 subjects.

### 2.12. Mechanosensory Conditioning

The two air flux speeds were 1.25 m/s and 5 m/s. At baseline, there was no air flowing; also, this experiment did not require suction. The number of trials was increased to 32, with 16 of each CS, to allow more time for learning. Since these were novel stimuli, the speed of conditioning was unknown; therefore, more trials were preferable to allow for the observation of slower or later effects. There were *n* = 12 subjects.

### 2.13. Generic Experimental Protocol

Bees were collected from outdoor colonies, usually around midmorning, and then anesthetized one by one on ice. Using small tweezers to grasp the thorax, the neck of the bee was inserted into the slot on the bee mount (Appendix A, lower inset). The head holder (Appendix A, upper inset) was used to push the head of the bee forward until the posterior edge of the head holder lined up with the back of the bee mount. A piece of sponge was placed behind the body such that the subject was confined and lightly restrained (Appendix A, right). The bee mount was then placed on a post of the revolver facing outward (Figure 1B). This procedure was repeated for all subjects. The bees were then allowed to fully recover from the anesthesia—at least one hour—and subject suitability was tested by checking the innate PER with a stimulation with sucrose at the antennae. Bees without a strong PER were released. All bees were then fed with approximately 3 μL of a 50% weight/weight sucrose solution, and the revolver was placed into the experimental setup for a rest period of at least one hour. Through the PER_Protocol.m file, the experiment proceeded as follows.

After initializing the USB NI DAQ port, the Arduino and other components were powered on. Then, the camera was initialized, and each bee holder was manually oriented such that, in the test position, the head is directly facing the feeder. Next, the cover was mounted, the air flow turned on (for removing excess odor), and the precise feeder position was adjusted to be close enough for feeding but far enough to eliminate the risk of collision. One of the feeding vessels (p1000 pipette tips cut to size) was filled with 25% sucrose solution, and the sponge was soaked with the same solution. A trimmed toothpick (for the antennae stimulation) was inserted such that it rests on the sponge and can rotate to pass over the subject’s head close enough to touch the antennae. The precise height of the toothpick was manually verified to pass closely over the head of the bee, without collision but ensuring contact with the antennae; given the high consistency between subjects (and across experiments, given the same hardware) this rarely needed adjustment. Once the protocol parameters were set, the automated experiment began, with the feeder and sponge refilled with sucrose solution as they depleted over time. Once the protocol was finished, subjects were marked with a colored dot on the thorax and released. Lastly the surfaces of the machine were cleaned with ethanol. The sponge was replaced after each day of experiments.

### 2.14. Statistical Analysis

The machine learning classifier provides, for each frame, a probability of proboscis extension which replaces the classical binary yes/no categorization. Statistical analysis can now be performed on this continuous variable, testing its dependence on the stimuli and its temporal dynamics. Both are analyzed with a two-way repeated measures ANOVA, with CS+/− group as the between-subject variable and trials as the within-subject variable. This provides total group effects and the interaction between groups and trials. Simple between-subject group effects are calculated for each trial and corrected by controlling the false discovery rate (FDR) via the Benjamini and Hochberg method. Learning- and experience-induced changes manifest themselves in simple within-subject effects in the individual groups.

## 3. Results

### 3.1. Olfactory Conditioning

The first experiments were classic olfactory conditioning paradigms, where one odor was rewarded with a sucrose solution, and another was not. In both cases, the movements and timing of the machine and its stimuli were the same; the only difference was the rotation of the feeder from the left (containing sucrose solution), or from the right (empty). Both of the odors had a neutral valence for honeybees, as evidenced by similar results regardless of which one was rewarded. The paired and unpaired trials were presented in a pseudo-random order over ten trials, such that each subject receives trial 1 before moving on to trial 2, resulting in an 8 min inter-trial interval for a single bee (Figure 2A).

Evidently, our automated system achieves the same success as previous manual or semi-automated setups. By the end of three rewarded trials, the majority of subjects (74%) had learned the association between the odor and the sucrose, as quantified by the extension of the proboscis to the odor presentation prior to the sucrose delivery. This was calculated using the CNN classifier: at every camera frame the classifier outputs a probability that the proboscis is extended, and we set a threshold (0.8) over which we categorize the frame as having the proboscis extended. We removed from the analysis any subjects which did not extend the proboscis to the antennal sucrose stimulation, since if they have lost their innate PER, we cannot use their PER as a behavioral metric for learning. We also removed subjects which already had the proboscis extended for greater than half of the second prior to the onset of the CS, to avoid classifying as “learned” those subjects which already had the proboscis extended (either by chance or due to associating the context with the reward) when the CS began. This procedure typically excluded 0.3–5% of responses over an entire experiment. With the remaining subjects, within the time window of the CS which did not overlap with the US, we classified a subject as having learned when the proboscis was extended for one quarter of a second or longer.

By the last paired trial, the PER was 83%. Meanwhile, the unpaired odor did not elicit the same learned association response, never reaching a higher percentage than 13% (Figure 2B). Statistical analysis shows a significant main effect between groups (*F*(1, 46) = 35.9, *p* = 3 × 10^−7^). Simple within-subject effects show learning in the CS+ group (*F*(4, 92) = 22.2, *p* = 7.3 × 10^−15^) but not in the CS− group (*F*(4, 92) = 0.68, *p* = 0.61). This group dependence of learning is also reflected in a significant interaction of group × trial (*F*(4, 184) = 19.3, *p* = 2.7 × 10^−13^). During the first trial, the difference between groups is not significant (*F*(1, 130) = 0.015, *p* = 0.91), but this difference already becomes significant at the 2nd trial (*F*(1, 130) = 4.93, *p* = 0.036) (Appendix A).

### 3.2. Mechanosensory Conditioning to a Low Air Flux

To further explore the stimulus possibilities with our setup, we ran a PER experiment using stimuli which were not two different odors but rather two different magnitudes of air flux (without odor). We consider this task to be more challenging because, rather than being distinguishable by chemical compounds, as in the case of odors, different air fluxes are the same quality of stimulus presented in different quantities. The protocol was the same as for the olfactory experiment, except for the type of stimulus and the number of trials (which was increased from 10 to 32 to allow more time for learning). Here, the lower flux was rewarded (CS+) and the higher flux was not rewarded (CS−). Notably, the classic PER learning curve (Figure 3A) is similar to a simple olfactory PER experiment: 83% of honeybees succeeded by the third CS+ trial, peaking at 92% on trials 5, 6, 10, and 14; meanwhile the response to the CS− was never higher than 17%.

In addition to the traditional categorical PER scoring, our setup allows for analysis of the response behavior in more detail, since the probability of proboscis extension is available at every camera frame (such analyses are available for the olfactory experiment in Appendix A). The average time curves across bees are shown in Figure 3C, where the CS period is shadowed yellow, and the sucrose reward period (provided only during paired trials) is shadowed magenta. A quantitative analysis of the time-resolved process before and during proboscis extension would not be possible with manual observation.

The time averaging over single trial periods provides a mean area under the probability curve (Figure 3B). For the CS+ group, this area is measured only prior to the arrival of the US, since the proboscis extension during the sucrose administration is very stereotyped and therefore does not provide any further information. However, the CS− group is not confounded by the US and, in fact, continues to show important features; so that area is analyzed across the entire trial. To render the integrated probabilities comparable, they are divided by time. See Appendix A for this analysis performed equally for the CS+ and CS− conditions, both before and after the arrival of the US.

Looking at this temporally resolved behavior, learning in the CS+ group is revealed by a trial-by-trial reduction in reaction time to the CS+ while the CS− reaction time remains high throughout the entire experiment (Figure 3D); additionally, there is an increased probability amplitude signaling greater proboscis extension overall (Figure 3C, red curves). The integration over trial periods shows that a full consolidation of a stereotyped response requires seven learning trials (Figure 3B, red curve), which is not obvious from the classical learning curve (Figure 3A, red curve). The CS− group shows hardly any response during the CS− stimulus, but after the stimulus is switched off, a curious tendency of proboscis extension is noticeable (Figure 3C, blue curves, magenta region). This effect increases, clearly visible after averaging over the trial periods, until trial 7, but then decreases again to its initial level after 13 trials (Figure 3B, blue curve).

The statistical analysis of the proboscis extension probability (per time) shows a significant main effect between groups (*F*(1, 22) = 12.0, *p* = 0.0023). Simple within-subject effects are significant in the CS+ group (*F*(15, 165) = 2.88, *p* = 4.8 × 10^−4^) and, different from the odor conditioning experiment, also in the CS− group (*F*(15, 165) = 3.96, *p* = 4.5 × 10^−6^). The trial-dependent changes still depend on the group, manifested in a significant interaction group × trial (*F*(15, 333) = 2.39, *p* = 0.0028). Simple between-subject effects become significant only on and after trial 7 (*F*(1, 85) = 5.30, *p* = 0.038) (Appendix A).

### 3.3. Mechanosensory Conditioning to a High Air Flux

In a third experiment, we inverted the rewarded and unrewarded conditioned stimuli. Now, the high air flux is associated with the sucrose reward, while the low flux is not. Both analysis methods, the classical binary evaluation of the proboscis extension (Figure 4A) as well as the PER extension probability curve (Figure 4B), show a similar picture. The unrewarded low flux (CS−) starts eliciting proboscis extension at a rapidly increasing rate arriving at 83% by trial 3. Compared to that, the rewarded high flux (CS+) shows a much slower increase arriving at 25% by trial 3. However, over time the CS− curve steadily decreases while the CS+ curve increases, until after 16 trials the situation has inverted to 25% for CS− and 67% for CS+. Though the CS− curve changes rapidly throughout the experiment, for the last few trials it is equal to or less than the CS+ curve (Figure 4A,B). The trial-by-trial temporal analyses (Figure 4D) show that the PER response latency steadily decreases for the CS+ throughout the experiment, although the response to the CS− remains faster until the final trial. The PER probability amplitude continues to change over the trials: it starts to increase for the CS+ group starting at trial 6 and reduces for the CS− group starting at trial 9 until the areas under these curves have inverted at trial 16 (Figure 4C).

Statistical analysis of the proboscis extension probabilities across subjects (Figure 4B) revealed a significant main effect between groups (*F*(1, 22) = 5.38, *p* = 0.031), although this was smaller than for the previous experiment where the low-flux was the CS+. Simple within-subject effects are significant in the CS+ group (*F*(15, 165) = 3.13, *p* = 1.7 × 10^−4^) and in the CS− group (*F*(15, 165) = 3.32, *p* = 7.2 × 10^−5^). The trial-dependent changes again depend on the group: the interaction group × trial is significant (*F*(15, 333) = 3.17, *p* = 7.1 × 10^−5^). Simple between-subject effects become significant starting at trial 3 (*F*(1, 85) = 8.89, *p* = 0.02) (Appendix A). The effect disappears again after trial 6, with one exception at trial 9 (*F*(1, 85) = 7.84, *p* = 0.02).

## 4. Discussion

Here we present a fully automated PER conditioning technique, from the mechanized stimulus delivery to the analysis with a neural network, which holds many advantages over the manual method. The delivery of stimuli can be precisely controlled both within and between subjects; an increased variety of stimuli is possible; and the code is easily customizable for individual needs. Furthermore, the internal validity becomes unquestionable when the entire protocol is automated and its parameters electronically saved, rather than relying on human performance and memory for administering the experiment and recording its data. Not only do these improvements allow for more easily standardized techniques, and therefore reliable comparison within and between labs, but they also reduce human error and bias during data analysis. Further, the precision of the control over the stimuli is crucial for maximizing the potential of the protocol. Of course, there is also the added bonus of saving researchers’ time: once the protocol code is running, the system requires no further manual input (except, for longer experiments, periodically refilling the feeder containing sucrose solution). Thus, in this work, we have begun to expand the limits of the PER method’s capabilities. With a fully automated system, experimental sessions are no longer constrained by the stamina of the researcher, and as a result can be multiple hours long, allowing us to propose difficult tasks and observe the bees’ behavioral changes over many trials. During our mechanosensation experiments, increasing the number of trials provided valuable information which would have been incomplete during shorter experiments, and only on the last couple of trials did we begin to see signs that the bees were becoming satiated, as indicated by slightly decreased AUC during the period of sucrose delivery (Figure 3C and Figure 4C).

Although video recordings of PER experiments have been used for data analysis for over two decades, they were always limited by the labor-intensive nature of manual scoring [34,35]; even efforts to automate insect body part tracking, though largely successful, still required substantial manual input, especially between multiple subjects [14,15,19]. Meanwhile, our real-time analysis with a neural network allows researchers to access instant results since the network analyzes video recordings throughout the experiment. Further, despite those early approaches providing more precise information by tracking individual body parts, they encountered severe problems when these body parts overlap: one study reported error rates of 0–22% in the PER classification for single subjects [19]. Meanwhile, our model reaches a classification accuracy of 99.7% within its training sets. Typically, when analyzing a new experiment, we will visually check some videos against the model’s output to ensure its accuracy; if there are any misclassifications, we retrain the model on videos from that experiment. However, this happens less and less frequently as the model is trained on data from different experiments and therefore becomes more robust.

More recently, DeepLabCut was used for the first time to track honeybee motion in an investigation of antennal motion patterns and odor valence [18]. For these experiments, the network was trained on multiple body parts for every single video, with few frames (3–10) required for each video. Our approach needed a far larger training set, a few hundred frames, for our convolutional neural network to reach near-perfect accuracy, but rather than individually labeled body parts it only requires a binary classification of whether or not the proboscis is extended in a given frame (a much quicker manual process). Further, thanks to the excellent reproducibility of the head position among subjects, this training is required once and then the trained model works for all subjects. So far, DeepLabCut has not been tested specifically on the proboscis, and it may be a less ideal tool for such a body part which is only visible some of the time. Our convolutional neural network provided the most efficient solution for a paradigm in which we only need to track a single body part.

Evidently, our automated system achieves the same success as previous manual or semi-automated setups. The first olfactory experiment, designed according to the recommended stimulus timing [13], produces a textbook learning acquisition curve wherein most subjects have successfully learned the association by the third rewarded trial. The peak success rate is comparable to that of previous studies [5], while there is very little response to the CS− (Figure 2B). Although simple, this experiment validates our methodology so that we may move on to exploring novel stimuli.

In an initial experiment we presented two different speeds of air flux at a ratio of 1:4, with the lower air flux equal to 1.25 m/s and the higher air flux equal to 5 m/s. This ratio was chosen according to animal research regarding numerosity, in particular Weber’s Law, which states that the ease of discriminating between two magnitudes depends on their ratio [36]. A lower ratio of 1:3 was tested and proved to be too difficult, with a very low rate of successful discrimination (see Appendix A). We did not test any higher ratios: increasing the ratio requires either a decreased low flux, which is limited by the technical capabilities of the valves, or an increased high flux, which could begin to depart from ecological relevance and become aversive (as we speculate below, we may already have an aversive element to our high air flux).

The results show a classical learning curve (Figure 3A) similar to that of the odor learning experiment (Figure 2A). The latency to proboscis extension reflects the same learning effect through a different measure (Figure 3D). Interestingly, the rate of acquisition and the overall success rate more closely mimic those of olfaction experiments rather than vision experiments, possibly due to the findings regarding mechanosensation processing in the antennal lobe as discussed above. Furthermore, the novel measure of the proboscis extension probability (Figure 3B), which can be considered a rating of the PER distinctiveness on a given frame, allows for the detection of minor responses that would normally be simply classified as “not extended”. This reveals slight responses to the CS− stimulus at the beginning of an experiment which disappear after a few trials, suggesting a certain degree of initial generalization between the CS+ and CS− stimuli. The incomplete extension suggests indecision or uncertainty during the response, which is remarkable because PER was previously considered a reflex that is strongly triggered whenever a learned stimulus is identified. Yet this proposition is in agreement with recent evidence indicating that mechanisms of attention may underlie associative learning in insects [37]. Our closer look at the behavioral process suggests that PER is a behavior containing large variability, which in part reflects the degree of certainty with which the decision was made regarding stimulus identity; such a measure is particularly useful at the beginning of the learning process when the animal’s certainty is low. This phenomenon would remain unnoticed if only full proboscis extensions were counted by applying a threshold to the PER probability (Figure 3A). The tendency to generalize between stimuli during PER conditioning was not found in the vast majority of previous studies; it is possible that such a phenomenon went undetected due to the lack of subthreshold scoring. However, in the work of Bitterman et al. [4] a similar small and fast decaying response to the CS− is present in odor conditioning experiments.

The time-resolved monitoring of the process (Figure 3C) gives further insights into the decision-making process: most of the subtle responses to the high flux CS− stimulus appear immediately after it is switched off. At that point, for a short moment, the decreasing air flux velocity equals the value of the lower flux CS+. If a subject has correctly learned the association with the low flux CS+, this moment might prompt the timid responses observed. Such temporal resolution will allow for the investigation of interesting aspects of decision making, such as speed–accuracy dependence [38], since we can now measure with high precision the onset of right and wrong responses. Moreover, the shape of the response curve may provide information regarding the coding mechanism in the neurons involved in the decision-making process. Experiments in vertebrates show complementary contributions of latency and intensity coding [39], while in insects both coding mechanisms could be identified, so far, only in the periphery of the olfactory system [40]. It will also allow for the study of the temporal evolution of decision making for stimuli that require temporal integration for their identification, like oscillations [41]. For honeybees, this is particularly interesting to address the open questions of how the waggle dance is decoded in the brain [22,26].

In the second mechanosensory experiment, the stimuli were switched such that the CS+ was the high flux and the CS− was the low flux. The resulting data were surprisingly different from the previous experiment; so much so, that it is possible the stimuli do not have equivalent valences as we originally thought. We suggest that the high air flux is not a neutral stimulus, but a mildly aversive one, due to the fact that honeybees take many more trials to learn its association with a reward than for the low air flux (Figure 4A). Similar results are seen for PER experiments using other stimuli which are known to have a biological valence, such as the sting alarm pheromone (SAP), where the main component of SAP (isoamyl acetate) impaired the bees’ learning of the sucrose association [42]. This effect may be because, in ecological settings, proboscis extension (typically for feeding or foraging behavior) and the presence of SAP should be mutually exclusive events. Thus, the bees need to learn to extend the proboscis to a stimulus which would normally inhibit such an action. This may be a factor for our results as well: if a honeybee would normally encounter such an air flux only while flying, it will be an ineffective stimulus for prompting the feeding-oriented proboscis extension. In other experiments, where the SAP component was directly used as a CS+ stimulus, the learning success was again strongly reduced and an unusual generalization could be observed: after conditioning to isoamyl acetate, bees exhibited PER for several novel odor stimuli with very different chemical properties [43]. A similar scenario was observed in the experiments presented here. While the response to the high flux CS+ was less than expected, there was an apparent generalization to the CS−, with the low flux evoking PER as strongly as if it was rewarded. Over time, additional trials slowly reversed this trend (Figure 4C,D). Interestingly, the opposite task—bees learning to withhold the proboscis despite being stimulated with sucrose—has already been successfully performed [44]. There is a separate protocol to study aversive learning by using the sting extension response (SER) in a similar way as PER, wherein a neutral CS is paired with an aversive US; we believe our work and those discussed here demonstrate that PER can provide complementary information about behavioral responses to aversive stimuli, with SER assessing the presence of defensive behavior and PER assessing the lack of feeding behavior.

These experiments call into question not only valence but also magnitude perception. By using different speeds of air flux as the CS+ and CS−, we pose a quantitative task rather than a qualitative one as in the case of odors. The effect of stimulus intensity has already been explored with PER in the olfactory domain [45,46] including an experiment in which honeybees succeeded at discriminating a high concentration CS+ from a low concentration CS− of the same odor but failed with the reverse discrimination [47]. This asymmetry was postulated to be due to differential stimulus salience, with the higher, more salient odor concentration facilitating learning more than the other. Here, instead, our high air flux (the more intense) stimulus was poorly received as a CS+ while the low air flux was readily learned. Our opposite result supports the idea that the high air flux is aversive, thereby overcoming the saliency effect; alternatively, it may indicate an implicit difference in modality-dependent intensity coding, or derive from ecological relevance (a stronger stimulus may not necessarily be more salient if it then becomes less relevant given the context—in this case, a honeybee might not be inclined to associate a high air flux, which may occur during fast flight, with simultaneous feeding behavior). Future mechanosensory experiments may assess changes in air speed rather than the presentation of absolute air speed values, since the former is more biologically relevant [21]. This could be accomplished by maintaining a constant moderate air flow, which would be increased or decreased for the CS+ and CS− before returning to the baseline air flow. In this way we could avoid complications seen here where the high air flux inevitably is briefly equal to the low air flux during the onset and offset. The technical precision of our setup will also provide an advantage to experiments using tactile CS, to investigate whether the intensity effects discussed here would generalize to another type of mechanosensation.

Future studies of complex learning and decision-making processes with conflicting stimuli will benefit from the high temporal resolution of individual responses. This provides a tool that allows for experiments which test the limits of insect cognition [48]. For example, one could test whether emotion-like states can have an influence on decision-making not only at a scale that changes the final outcome, as shown in a previous study [49], but also on a more subtle level wherein only the dynamics of the process may be altered. Adding other distractive stimuli would allow for studies on selective attention [50]. Additionally, studies on brain lateralization may benefit from an automated analysis with high temporal resolution. PER conditioning has contributed substantially to the discovery and characterization of left–right asymmetries in the honey bee brain across olfactory as well as gustatory modalities [51,52,53], but these projects require two experiments with one of the antennae inactivated with either a silicon coverage or a physical blockage. We propose a more direct approach of providing stimuli from either the left or the right side while both antennae remain active. In summary, our flexible automated setup has revealed novel insights into the temporal dynamics of behavioral responses to a novel CS; and it will allow researchers to expand their experimental repertoire to ask more subtle, complex questions, to peer more deeply into honeybees’ learning and decision-making processes.

## Figures and Tables

**Figure 1 insects-15-00094-f001:**
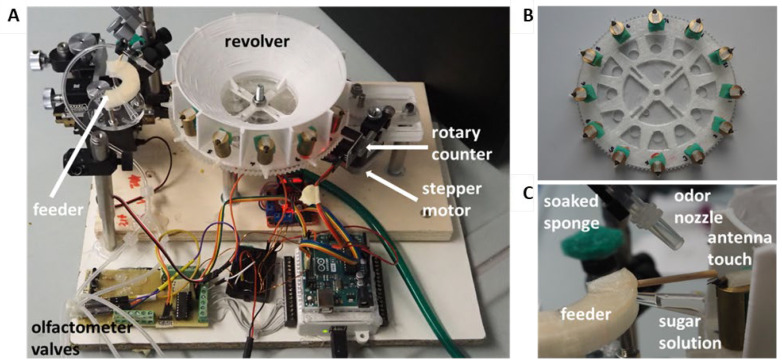
Automated setup characterization. (**A**) Full setup showing the main elements: revolver, feeder, stepper motor and rotary counter, olfactometer, and control electronics. (**B**) The revolver in isolation with 12 bees mounted around the circumference. (**C**) Feeder details including a sucrose-soaked stick to touch the antennae, an olfactometer output nozzle, and a sucrose-soaked sponge on which the stick rests.

**Figure 2 insects-15-00094-f002:**
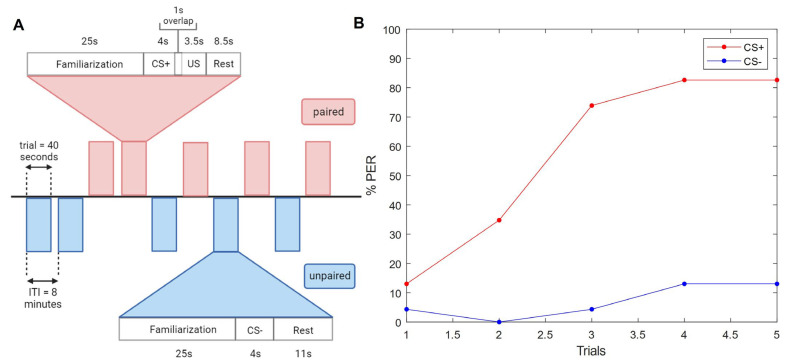
A proof-of-concept demonstration of an olfactory conditioning experiment with the automated system. (**A**) A protocol diagram representing one complete experiment with 10 pseudo-randomized trials. The timelines are depicted for one trial of each of the paired and unpaired paradigms. One trial contains the time between the bee moving into the test position and moving out of it. The inter-trial interval (ITI) is the amount of time between one trial beginning and the next trial beginning, for one individual bee. This is also indicative of one complete rotation of the wheel holding the 12 bees. (**B**) Classic PER response curve over trials. *n* = 24. 1-hexanol (red) was rewarded (CS+) and 1-nonanol (blue) was not (CS−).

**Figure 3 insects-15-00094-f003:**
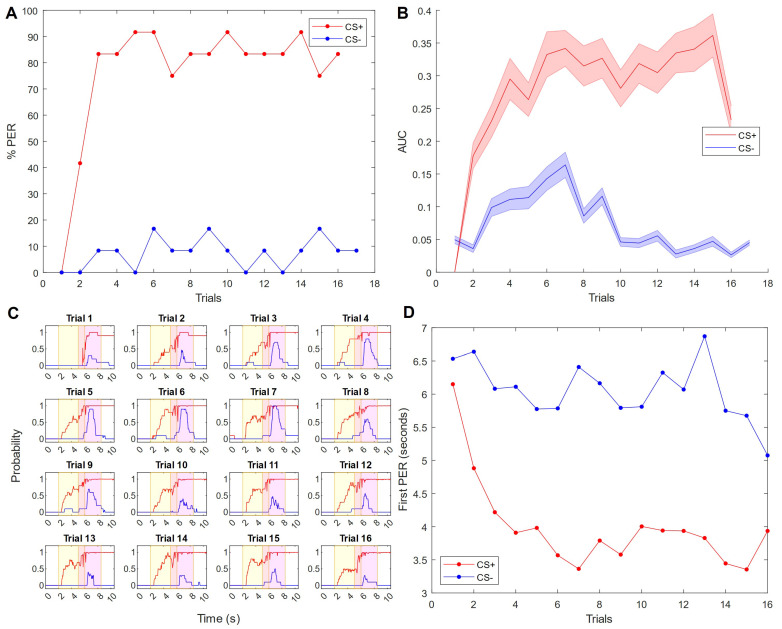
Honeybees successfully perform PER with two different (odorless) air fluxes, where the lower flux is rewarded. (**A**) Classic PER response curve over trials. The low air flux stimulus (1.25 m/s), in red, was rewarded (CS+). The high air flux stimulus (5 m/s), in blue, was not rewarded (CS−). (**B**) Mean area under the probability curves for CS+ and CS− stimuli for each trial. This is a measure of the probability over time, as given by the classifier, that a bee has the proboscis extended. (**C**) A trial-by-trial view of the results in (**B**), where the probability of the proboscis extension is plotted over time in seconds. Results are averaged across all bees (*n* = 12). The yellow panel indicates the time duration of the conditioned stimulus (air flow) and the magenta panel is the time duration of the unconditioned stimulus (25% sucrose solution). There is a 1 s overlap. (**D**) The mean proboscis extension latency of subjects which respond during a given trial, in seconds.

**Figure 4 insects-15-00094-f004:**
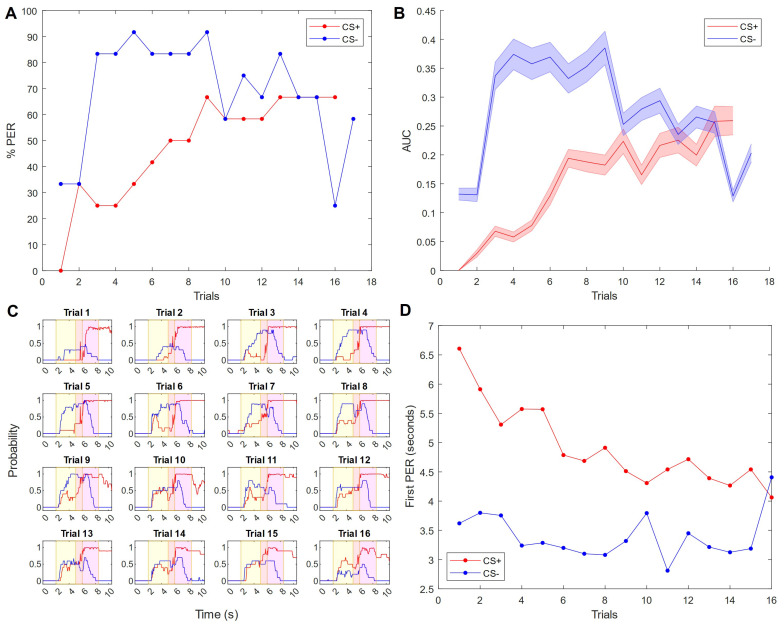
Honeybees are less successful with two different (odorless) air fluxes, where the higher flux is rewarded. (**A**) Classic PER response curve over trials. The high air flux stimulus (5 m/s), in red, is rewarded (CS+). The low air flux stimulus (1.25 m/s), in blue, is non-rewarded (CS−). (**B**) Mean area under the probability curves for CS+ and CS− stimuli during each trial. This is a measure for the probability over time, as provided with the classifier, that a bee has the proboscis extended. (**C**) A trial-by-trial view of the results in (**B**), where the probability of the proboscis extension is plotted over time in seconds. Results are averaged across all bees (*n* = 12). The yellow panel indicates the time duration of the conditioned stimulus (air flow) and the magenta panel is the time duration of the unconditioned stimulus (25% sucrose solution). There is a 1 s overlap. (**D**) The mean proboscis extension latency of subjects which respond during a given trial, in seconds.

## Data Availability

With the exception of the neural network and the video recordings, all data supporting the published results (including experimental data, software for experimental control and analysis, technical drawings of mechanical parts, and electrical circuits) are publicly available in the following repository: https://github.com/NeurophysicsTrento/Automized-PER (accessed on 22 December 2023).

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
