# Peer review of "Associative Learning of Quantitative Mechanosensory Stimuli in Honeybees"

_insects, 2024, doi:10.3390/insects15020094_

Round 1
Reviewer 1 Report
Comments and Suggestions for Authors
The manuscript by Strelevitz et al presents a novel, automatized method to record the proboscis extension response (PER) of honey bees in Pavlovian experiments in which a conditioned stimulus (CS: an odorant or an air flow, in this case) is paired with a reward of sucrose solution (unconditioned stimulus or US). The method is fully automatized, including CS and US delivery, so that compared to recent methods which attempted to provide a finer quantification of PER, the advantages of this novel methodology are significant.
The manuscript has three mains goals: 1) to introduce the novel method and its components – hardware and software – which are rendered accessible via open-source platforms, 2) to apply the method to the analysis of an olfactory discrimination learning as a proof of concept, and 3) to test the suitability of a novel version of PER conditioning in which air flows with different speeds serve as CS.
The manuscript is relevant from the methodological point view and represents a significant advancement for the study of honey bee olfactory learning. It is extremely important and commendable that the authors have given free access to all components of their recording method, so that anyone can use it in the future. I have some general comments that can help clarifying certain points but overall, this is a very good work.
General comments
1) The Introduction omits a series of studies in which mechanosensory antennal information was used to condition harnessed bees. This is a pity as it creates the impression of an uninformed work presentation. One work is cited in the Discussion1, but it should be mentioned already in the Introduction in order to provide a fair view of mechanosensory conditioning in bees. Other works have used mechanosensory stimulations and PER but in an operant framework (i.e. the bees had to touch actively the stimuli proposed to them) to access the reward2-5. It would be interesting to mention these works too, as the method developed by the authors could be useful also for this different context.
2) I would appreciate more details concerning US delivery. This was for me a point of interrogation. The antennae of a harnessed bee are mobile and in manual experiments one has to repeatedly touch them in order to ensure a good contact with the sucrose solution and full extension of the proboscis. If the stick soaked in sucrose solution is moved towards the bee in an automatized way, full contact may not be ensured in some cases, thus affecting the possibility of recording PER when one should have occurred. How did the authors control this point?
3) The asymmetry found in discrimination success when using air flows of different speed reminds results obtained by Pelz et al6 when using two concentrations of the same odorant as CS+ and CS- in a similar differential conditioning assay. These results are incorrectly described in the Discussion as a case in which bees were unable to learn the difference between the two concentrations of the same odorant. This is wrong as in this case, bees learned efficiently to discriminate the high concentration as CS+ from the low concentration as CS-, yet the reversed discrimination was not possible. These results are interesting as they seem to have a different basis when compared to the ones reported in the manuscript where the high airflow speed could hardly be learned as CS+ while the low airflow speed could be learned efficiently. In the case of odorants, a plausible explanation would refer to stimulus salience, which is crucial variable in classical conditioning7. Learning was facilitated in the case of the higher odorant concentration given its higher salience. When the CS- had the higher concentration, learning was impaired as it attracted more responses than the CS+ with a lower concentration. Salience does not apply to the asymmetry detected for the airflow speeds, as the results predicted would be the opposite of what was obtained. This reaffirms the explanation provided by the authors in terms of the aversive nature of the high airflow speed. Discussing this point could strengthen this last interpretation.
Minor comments
Line 192: The expression “be aware” does not fit with the general style of the manuscript. The sentence could be rewritten as: “The appropriate voltage required by the valve was, in our case, 12V for activation …”
Line 277: Explain why these two odorants were chosen.
Line 279-282: You mention that conditioning performance was the same irrespectively of which odorant was used as CS+ and as CS-. Your results (Fig. 2) show a case in which hexanol was the CS+ and nonanol the CS-. Why didn’t you show the reversed case, to make the contrast with the air flows even clearer?
Line 283: 25% (w/w)? Specify.
Figure 1: You have used the term “habituation” to describe the first 25s of a trial; this is an unhappy choice in a paper on associative learning as habituation means something very different in learning theory and refers to a form of non-associative learning. Replace by “familiarization”.
Lines 370-371: these individuals may have reacted not by chance – as stated here – but because they associated the context itself with reward.
Lines 614-615: The sentence on the sting extension response has no prior explanation so that it cannot be fully understood. You have spoken about the aversive nature of stimuli and its impact on PER responses but never mentioned that there are protocols using SER to study aversive learning. Include mention to this fact in rder to make your sentence understandable.
Line 620-621: The account of the experiment by Pelz et al6 is incorrect. The bees learned to discriminate high from low intensity of the same odorant but did not learn the reversed discrimination. This point is developed above. Please describe the work by Pelz et al appropriately.
Line 637: You may want to add a reference to the work by Baracchi et al as it expands your arguments to the gustatory modality8.
References
1 Giurfa, M. & Malun, D. Associative mechanosensory conditioning of the proboscis extension reflex in honeybees. Learn Mem 11, 294-302, doi:10.1101/lm.63604 (2004).
2 Scheiner, R., Kuritz-Kaiser, A., Menzel, R. & Erber, J. Sensory responsiveness and the effects of equal subjective rewards on tactile learning and memory of honeybees. Learn Mem 12, 626-635, doi:10.1101/lm.98105 (2005).
3 Kisch, J. & Erber, J. Operant conditioning of antennal movements in the honey bee. Behavioural Brain Research 99, 93-102 (1999).
4 Erber, J., Pribbenow, B., Grandy, K. & Kierzek, S. Tactile motor learning in the antennal system of the honeybee (Apis mellifera L.). Journal of Comparative Physiology a-Sensory Neural and Behavioral Physiology 181, 355-365, doi:DOI 10.1007/s003590050121 (1997).
5 Pribbenow, B. & Erber, J. Modulation of Antennal Scanning in the Honeybee by Sucrose Stimuli, Serotonin, and Octopamine: Behavior and Electrophysiology. Neurobiology of Learning and Memory 66, 109-120, doi:https://doi.org/10.1006/nlme.1996.0052 (1996).
6 Pelz, C., Gerber, B. & Menzel, R. Odorant intensity as a determinant for olfactory conditioning in honeybees: Roles in discrimination, overshadowing and memory consolidation. J Exp Biol 200, 837-847 (1997).
7 Rescorla, R. A. & Wagner, A. R. in Classical conditioning II: Current research and theory (eds A. H. Black & W. F. Prokasy) 64-99 (Appleton-Century-Crofts, 1972).
8 Baracchi, D., Rigosi, E., de Brito Sanchez, G. & Giurfa, M. Lateralization of Sucrose Responsiveness and Non-associative Learning in Honeybees. Front. Psychol. 9, 425, doi:10.3389/fpsyg.2018.00425 (2018).
Author Response
Please find our responses in the attached file.

Reviewer 2 Report
Comments and Suggestions for Authors
In this manuscript, the authors constructed an automated apparatus for classical PER conditioning using honeybees. This new training method is useful because it can exclude human error and more importantly, as described in the text, it allows qualitative analysis of the PER (as in Fig. 2BC, 3BC). The results are clear and well-described. I only have some minor comments as follows.
1) As far as I know, there have been some reports for semi-automated PER conditioning apparatus. It would be better to describe them and what the exact improvements in this study are.
2) Fig. 4 should be Fig. 1? Along with, Fig. 1, 2, 3 should be Fig. 2, 3, 4, respectively?
3) In Fig. 4C and Fig. S1D, I could not fully understand how the bee was fed sucrose solution. Eventually, I understood by watching the supplementary video S1, but I think it would be better to explain more clearly in the Materials and Methods, somewhere around L176.
4) Is ‘schematic circuit (L147)’ Fig. S1A?
5) Descriptions about how the bee was mounted were partly duplicated (L165-169 and L303-309). Please simplify the text.
6) Bhagavan and Smith (1997) reported that honey bees could discriminate the same odorant with different intensities. In this case, the odor with a higher concentration elicited less generalization compared with that with a lower concentration, which was the opposite effect in the present study, the airflow with a lower speed elicited less generalization. This might be caused by modality-dependent intensity coding as the authors described (L621).
Bhagavan S, Smith BH (1997) Physiol Behav 61: 107-117
7) Finally, I could not fully understand the biological meaning of the association of the airflow speed with PER for the honeybee. Please explain somewhere in the Introduction or Discussion.
Author Response

(The authors gave the same response as above.)

Reviewer 3 Report
Comments and Suggestions for Authors
This manuscript presents a fully automatic setup to train honeybees in associative learning tasks and to analyse the data. Such a setup improves both replicability of experimental procedures and the grain of the behavioural output recorded. Although several labs have developed over time tools and experimental procedures to increases the quality of PER experiments, I appreciate the great advantages of having developed such a setup and this development certainly deserves publication in a methodological paper in particular as both information about the hardware and the software are made available for replication in other labs. However, the learning experiments provided are not satisfactory in making conclusion on the ability of bees to learn quantitative mechanosensory stimuli as claimed in the title. One major concern is the sample size (N=12 bees for 2 groups, so N= 6 bees by group ??) which is far too low by comparison to standards in the field and could be even considered ridiculous in particular when using an automatized setup. Those 12 (or 24 bees if it was 12 bees by group) means only one day of experiment. I admit that I don’t understand, given the effort invested in developing the setup, that no more data were collected to increase statistical robustness of the results and propose more experiments to further describe quantitative mechanosensory stimuli learning in bees. Some ideas of follow-up experiments: could bees also learn to detect different intensity of tactile stimuli if provided with an object rather than an air flow or in an other part of the body ? Could you confirm the generalisation occurring when trained with a relatively intense stimulus by comparison that when using a less intense one ?
Other comments (following the manuscript order):
L65: Quoting only an early attempt to perform visual PER in bees does not reflect the state-of-the-art. Please add more references or quote a review addressing in details the successes and difficulties faced when using PER conditioning with visual stimuli: Avargues-Weber and Mota 2016 doi:10.1016/j.jphysparis.2016.12.006.
l.67 to 70. I think this is not a fair description of standard protocols used in the field. Most labs use olfactometers and electronic devices to control for olfactory stimuli flux and timing for example.
Introduction: A review on PER conditioning using mechanosensory stimuli should be provided.
l.136: This procedure to select foragers implies that only pollen foragers were recruited for the experiment which is known to have an impact on learning abilities differing from those of nectar foragers (see works from Scheiner, Page and Erber for example). This should be discussed.
l. 283: Could you provide the concentration of the sucrose solution in mol/L please to facilitate comparison between studies. Was it a percentage by volume or by weight ? In any case, this seems a lower concentration from the usual concentration used to reward bees in learning experiment. Could you justify this choice ?
l. 301: 12 subjects in total (only one run of the experiment as the setup allows to train 12 bees in parallel if I understood well ??) or 12 by group ?
l.323: You used a sponge filled with sucrose solution to reward the bees. How did you avoid bacterial development ? Did you change the sponge for each experiment ?
l. 366-370: Which % of subjects were removed following this procedure ?
l.417: You analysed the proboscis extension for 6 sec in CS- trials by opposition to the first 3sec (when the CS is provided alone) in CS+ trials. I know you divided the number of events by 2 to compensate but I would recommend to analyse the first 3 sec and the last 3 seconds separately as bees may behave differently and this could be interesting per se.
l.546: Interesting information, why not showing the data ? Could it be that there were less than 12 individuals involved in the experiment ??
l.556-575: Analysing incomplete proboscis extensions are indeed a great addition to classical PER studies offered by your setup. However, I regret that you did not apply this level of analysis to your olfactory PER experiments to see if it could also add valuable information nor compared the % of incomplete PER and full PER to analyse if both values correlate which would suggest that incorporating incomplete PER would increase the sensitivity of learning detection or whether you observe a different pattern being thus potentially informative about level of certainty in the bee’s perspective. Both cases are interesting but they won’t have the same consequence in term of data interpretation and questions for which it would be useful to add this level of analysis.
Author Response

(The authors gave the same response as above.)

Round 2
Reviewer 3 Report
Comments and Suggestions for Authors
The authors have addressed most of my comments in a satisfactory way. I only regret that they do not present all data collected: they mentioned in their response letter 'We conducted a series of experiments, varying several CS and US parameters. We report here the result that yielded the best learning performance.' I consider that those information on what did not work that well is certainly also of high interest for readers and has implication on the physiology underneath. I strongly recommend adding those experiments in the manuscript.
Author Response
We hope to have satisfied the reviewer's request with the following changes:
We have added two data sets of imperfect learning experiments, where the ratio between CS+ and CS- was too small, as supplementary Figures S4 and S5, and cite them in the main test line 560.
We now mention another crucial parameter that led to imperfect learning, the familiarization time, in the text from line 279.
Round 3
Reviewer 3 Report
Comments and Suggestions for Authors
Thank you for this addition.
Author Response
You are welcome